# Usefulness of Endoscopic Retrograde Cholangiopancreatography in the Diagnosis and Treatment of Traumatic Pancreatic Injury in Children

**DOI:** 10.3390/diagnostics13122044

**Published:** 2023-06-13

**Authors:** Seong Chan Gong, Sanghyun An, In Sik Shin, Pil Young Jung

**Affiliations:** Department of Surgery, Yonsei University Wonju College of Medicine, Wonju 26426, Republic of Korea; surgeon_g@yonsei.ac.kr (S.C.G.); uldura@yonsei.ac.kr (S.A.); fleece2@yonsei.ac.kr (I.S.S.)

**Keywords:** cholangiopancreatography, endoscopic retrograde, pancreas, injuries, pediatrics, diagnosis

## Abstract

Pediatric trauma patients are increasing, and trauma is the leading cause of death in children. Pancreatic injury is known as the fourth most common solid organ injury, but the diagnosis of pancreatic injury is often delayed due to the retroperitoneal location of the pancreas and the low sensitivity and specificity of diagnostic tests. Endoscopic retrograde cholangiopancreatography (ERCP) is an important test for the diagnosis and treatment of various biliary tract and pancreatic diseases. However, cases of performing ERCP in traumatic pancreatic injury in children have been rarely reported. Thus, we aimed to evaluate the usefulness of ERCP in traumatic pancreatic injury in children. Between January 1983 and December 2022, pediatric patients under the age of 18 who were treated for traumatic pancreatic injury at a single institution were recruited and retrospectively analyzed. Patient characteristics and clinical outcomes were assessed. Thirty-one patients were enrolled in this study. Among them, 15 (48.4%) patients underwent ERCP. The time to diet was significantly longer in the ERCP group. There were no statistically significant differences in other characteristics between the ERCP and the non-ERCP group. In nine (60%) patients of the ERCP group, ERCP was used for therapeutic intervention or as a decision-making tool for surgery, and was used to resolve pancreas-related complications. ERCP may be useful for the diagnosis and treatment of traumatic pancreatic injury in children. In addition, ERCP can be safely applied in children, and complications related to ERCP also may not increase. When obscure pancreatic injury is suspected, it is necessary to consider performing ERCP.

## 1. Introduction

Pediatric trauma patients are increasing, coinciding with the overall rise in trauma cases. Notably, trauma stands as the primary cause of mortality among children. Despite its rarity in children, pancreatic injury is recognized as the fourth-most-prevalent solid organ injury, trailing behind the spleen, liver, and kidney [1]. Among pediatric cases, blunt trauma is more frequently encountered compared to penetrating injuries, with reported incidence rates of 3–12% and 1.1%, respectively [2,3].

The pancreas is located in the retroperitoneum, making initial clinical symptoms and physical examination findings not evident in many cases; also, the specificity and sensitivity of tests such as serum amylase or lipase are low. Thus, an early diagnosis of pancreatic injury is often delayed [4]. Although pancreatic injury is not very common, a delay in early proper diagnosis is associated with high morbidity and mortality rates, with morbidity rates of 30–40% and mortality rates of 9–34% [5].

Most pancreatic injuries in children are caused by traffic accidents, child abuse, and bicycle handlebars damage [5]. Pancreatic injury occurs mainly due to the pressure of the pancreas between the spinal column and the abdominal wall. Unlike adults, children are more susceptible to traumatic pancreatic injury because they have flatter diaphragms, thinner abdominal walls, and higher costal margins [6].

Endoscopic retrograde cholangiopancreatography (ERCP) is a specialized procedure that combines endoscopy and radiological fluoroscopy techniques and is an important test for the diagnosis and treatment of various biliary tract and pancreatic diseases [7,8,9]. ERCP has been applied for more than 20 years in adults and with abundant experience, but cases of performing this procedure in children, such as for sclerosing cholangitis, biliary stones, pancreas divisum, biliary atresia, choledochal cyst, anomalous union of pancreatico-biliary duct (AUBPD), and traumatic pancreatic injuries, have been rarely reported. However, the frequency of performing has been steadily increasing [8,9,10].

As of now, there is a limited number of reports regarding the usefulness and safety of ERCP in traumatic pancreatic injury in children. Consequently, the purpose of this study was to assess the usefulness and safety of ERCP in traumatic pancreatic injury among pediatric patients. 

## 2. Materials and Methods

### 2.1. Patient Selection and Data Collection

This study was approved by the institutional review board of Wonju Severance Christian Hospital (IRB no. CR320181). Between January 1983 and December 2022, pediatric patients under the age of 18 who were treated for the traumatic pancreatic injury at Wonju Severance Christian Hospital were recruited.

The patients were initially evaluated through physical examination, laboratory tests, and imaging studies such as ultrasonography (US) or computed tomography (CT). Treatment decisions were generally based on the pancreatic injury grade according to the initial radiological diagnosis. However, the final treatment approach was ultimately determined by the surgeon’s clinical judgment (Figure 1).

The demographics and clinical characteristics of the patients, such as age, sex, body weight, initial blood pressure, initial shock status, injury mechanism, presence of abdominal co-injury, Injury Severity Score (ISS) and Pediatric Trauma Score (PTS), initial laboratory tests including serum amylase, lipase, C-reactive protein (CRP) and delta neutrophil index (DNI), and pancreatic injury site, whether CT, US or ERCP had been performed, pancreatic injury grade, early and late complications, the time to diet and the time to parenteral nutrition, intensive care unit (ICU) and hospital stay days, and mortality, were retrospectively reviewed.

Pancreatic injuries were graded according to the American Association for the Surgery of Trauma (AAST) guidelines (Table 1) [11] using all available clinical data, such as radiologic imaging, operative findings, and ERCP reports. Injury severities of patients were scored according to the ISS and the PTS. Early complications and late complications were defined as those that occurred related to pancreatic injury within 30 days after admission and those that occurred related to pancreatic injury 30 days after admission, respectively. Additionally, post-pancreatitis was defined as pancreatic abdominal pain with an elevation in serum amylase or lipase of more than three times the upper normal limit after ERCP.

The indications for surgery in this study were in the cases of hemodynamic instability with accompanying abdominal injury, complications requiring surgery, and grade III or higher pancreatic injury with the surgeon deciding that surgery was necessary.

### 2.2. The Indications and Procedure of ERCP

ERCP was indicated for both therapeutic and diagnostic purposes. Therapeutically, it was performed in cases where there was a suspicion of pancreatic duct injury based on radiologic imaging, such as CT and US. Diagnostically, it was used when there was uncertainty regarding the integrity of the pancreatic duct observed in radiologic imaging, or in cases of clinically suspected pancreatic injury.

During the study period, ERCP was performed by 3 gastroenterologists with extensive experience in adult ERCP. The procedure was performed with the same method as for adult ERCP in the patient’s left lateral decubitus or prone position. For the procedure, intravenous anesthetics such as midazolam, propofol, ketamine, and thiopental were used alone or in combination, and additional doses were used depending on the patient’s sedative status. 

ERCP was performed with a conventional side-viewing duodenoscope (TJF-260V, Olympus Corp., Tokyo, Japan). Endoscopic pancreatic sphincterotomy (EPST) was most often performed with a pull-type sphincterotome, but a needle-knife sphincterotome was used as needed. The size of the pancreatic stent used was 3 Fr, 5 Fr, or 7 Fr (Geenen^®^ Pancreatic Stent, Cook Medical, Bloomington, IN, USA/Advanix™ Pancreatic Stent, Boston Scientific, Marlborough, MA, USA).

### 2.3. Statistical Analysis

Continuous variables are presented in the mean ± standard deviation and the comparison between the two groups was analyzed by either the Student’s *t*-test or the Mann–Whitney U test. Categorical variables were analyzed using the Chi-square test and the Fisher’s exact test. The Wilcoxon signed rank test was used for nonparametric variables. All data management and statistical analyses were performed using SPSS version 26.0 (IBM, Armonk, NY, USA) and the values of *p* < 0.05 were considered statistically significant.

## 3. Results

### 3.1. Baseline Clinical Characteristics

Thirty-one pediatric patients with traumatic pancreatic injury were enrolled in this study. The mean age was 11.71 ± 4.86 years, and 21 (67.7%) patients were male. The average body weight was 42.18 ± 18.96 kg, and the number of patients who had initial shock was six (16.1%). The most common injury mechanism was blunt force (41.9%), followed by falling down (16.1%), motorcycle accident (12.9%), pedestrian traffic accident (9.7%), passenger traffic accident (9.7%), stab injury (6.5%), and unknown (3.2%). In particular, in the cases of blunt force, nine (69.2%) patients were injured by bicycle handlebars. To investigate the differences in injury mechanisms based on age groups, the patient population was divided into two groups: those aged 10 years and younger, and those older than 10 years. In the group of patients aged 10 years and younger, it was observed that the most common injury mechanism was blunt force (50%), followed by passenger traffic accidents (25%), pedestrian traffic accidents (16.7%), and falling down (8.3%). In the group of patients older than 10 years, it was found that blunt force remained the most common injury mechanism, accounting for 36.8%, followed by motorcycle traffic accidents and falling down (21.1% each) and stab injury (10.5%), while pedestrian traffic accidents and unknown were reported in 5.3% and 5.2% of cases, respectively (Figure 2). The number of patients with abdominal co-injury was 16 (51.6%), and the mean ISS and PTS were 27.32 ± 10.99 and 9.23 ± 1.88, respectively. The values of initial amylase and lipase were 582.69 ± 673.00 U/L and 3375.55 ± 6662.45 U/L, respectively, and CRP and DNI were 2.22 ± 3.72 mg/dL and 1.93 ± 2.24 %, respectively. The most common pancreatic injury site was the body, with ten cases (32.3%), followed by tail, head, and neck with eight (25.8%), five (16.1%), and three (9.7%), respectively. In addition, five (16.1%) patients had two or more injury sites. Among the patients, 29 (93.5%) had a CT scan and only 16 (51.6%) underwent US. Additionally, there were 15 (48.4%) patients who underwent ERCP, and 10 (32.3%) patients who underwent operation. In the final injury grade, grade II was the most common, with eleven cases (35.5%), grade I had six (19.4%), grade III and IV had seven (22.6%) each, and there were none with grade V. The numbers of patients with early complications and late complications were fifteen (48.4%) and six (19.4%), respectively. The time to diet and the time to parenteral nutrition were 9.04 ± 5.55 and 2.32 ± 1.57 days, respectively, and ICU stay days and hospital stay days were 2.65 ± 3.42 and 30.97 ± 59.41 days, respectively. Two (6.5%) patients died (Table 2).

### 3.2. Comparison between Operation and Non-Operation Group

The mean ISS was significantly higher in the operation group (35.20 ± 9.56 vs. 23.57 ± 9.70, *p* = 0.004). However, the mean PTS was significantly lower in the operation group (7.90 ± 2.13 vs. 9.86 ± 1.39, *p* = 0.005). Additionally, ICU stay days were significantly longer in the operation group (4.60 ± 3.84 vs. 1.71 ± 2.85 days, *p* = 0.025). The operation group tended to have more cases of abdominal co-injury (80.0 vs. 38.1%). Among the four patients with Grade II pancreatic injuries who underwent surgery, three had concurrent abdominal injuries (inferior vena cava rupture, liver laceration with kidney injury, and duodenal hematoma), while the remaining one underwent surgery based on the surgeon’s clinical judgment due to an initial elevation in serum pancreatic enzymes (Table 3).

### 3.3. Comparison between ERCP and Non-ERCP Groups

The time to diet was significantly longer in the ERCP group (10.93 ± 5.89 vs. 6.67 ± 4.21 days, *p* = 0.045). There were no statistically significant differences in other characteristics between the ERCP and the non-ERCP groups (Table 4).

### 3.4. Clinical Characteristics of Patients Who Underwent ERCP

Fifteen (48.4%) of the thirty-one patients underwent ERCP. The patients were 6–18 years old, with eleven males and four females. Among the injury mechanisms, seven were blunt force, all of which were bicycle-related accidents, and six of them were injured by bicycle handlebars. This was followed by two falls, two motorcycle accidents, one pedestrian traffic accident, one passenger traffic accident, one stab injury, and one unknown.

Early ERCP was defined as ERCP performed within 7 days after injury, and late ERCP was defined as ERCP performed 8 or more days after injury. The numbers of patients with early ERCP and late ERCP were twelve and three, respectively. 

Among the twelve patients who underwent early ERCP, therapeutic intervention was performed in four patients (patients 5, 8, 9, and 11) as a result of a grade 3 or higher injury. Although one patient (patient 14) was confirmed as having an injury of grade 2, the intervention was performed to help the secretion of pancreatic juice, and one patient (patient 13) was operated on for pancreatic duct injury. In six patients (patients 1, 3, 4, 6, 7, and 15), ERCP was performed for diagnostic purposes and it was identified as a grade 2 or less injury and treated conservatively.

Among the three patients who underwent late ERCP, two patients underwent the procedure postoperatively. One patient (patient 2) was admitted for stab injury. During the surgery, bleeder ligation and primary repair were performed due to liver laceration and stomach perforation, and no definite pancreatic injury was observed. During the follow-up, the serum pancreatic enzyme was elevated and a CT scan was evaluated, through which the laceration of the pancreas body and peripancreatic fluid collection were identified, and ERCP was performed on postoperative day 11. From the ERCP finding, irregular ductal change on the body of the pancreas without definite ductal leakage was observed. EPST and Endoscopic retrograde pancreatic drainage (ERPD) were performed. After that, the patient improved and was discharged. The other patient (patient 10) was admitted for unknown reasons (found lost on the road). Initial CT scans confirmed the presence of pancreas head transection, liver laceration, and spleen injury in the patient. During follow-up, pancreatic fistula with a drainage of pancreatic juice of amylase >6500 U/L through a drainage tube was identified, and ERCP was performed at postoperative day 15. On the ERCP finding, stricture of the neck of the pancreas was confirmed, but ductal leakage could not be clearly confirmed due to insufficient contrast day injection. EPST and ERPD were performed. After that, the patient improved and was discharged.

Patient 12 was treated conservatively as there was no definite pancreatic injury other than a mild elevation of serum amylase in the initial evaluation. During follow-up, the serum pancreatic enzyme was elevated and a CT scan was performed. Pancreatic pseudocyst was identified on the CT and percutaneous abscess drainage (PAD) was performed. After that, the patient improved and was discharged.

Of fifteen patients, early complications such as post-pancreatitis and pancreatic pseudocyst occurred in ten patients (patient 1, 2, 5, 8, 9, 10, 11, 12, 14, and 15) and late complications, such as chronic pancreatitis, pancreatic atrophy, and diabetes mellitus occurred in four patients (patient 4, 9, 10, 13) (Table 5).

### 3.5. Comparison of Radiologic, ERCP, and Final Injury Grade

When comparing the radiologic injury grade and the final injury grade with the Wilcoxon signed rank test, the final injury grade was significantly higher and the *p*-value was <0.001. When comparing the ERCP injury grade and the final injury grade, there was no statistically significant difference with a *p*-value of 0.190 (Figure 3).

## 4. Discussion

With the advancement of medicine and improvements in hygiene, deaths from infection or nutritional disorders have significantly decreased, while deaths from unexpected accidents have increased, relatively, making accidents the main cause of death in children [12,13]. In addition, as society develops, not only has the number of adult trauma patients increased, but also the number of pediatric trauma patients is increasing, and the causes of pediatric trauma are gradually diversifying [14].

Pancreatic injury is reported in 3–12% of blunt abdominal trauma and 1.1% of penetrating injury, and is known as the fourth-most-common abdominal solid organ injury after injury to the spleen, liver, or kidney [1,3]. Although pancreatic injuries are not common, children are more susceptible to traumatic injuries as they have flatter diaphragms, thinner abdominal walls, and higher costal margins, unlike adults [6].

Since the pancreas is located in the retroperitoneum, initial clinical symptoms and physical examination findings are often not clear. Along with this, the low sensitivity and specificity of the serum pancreatic enzyme test results in frequently delayed diagnoses of pancreatic injury [4]. For that reason, the incidence of pancreatic injury may be underestimated compared to the actual occurrence in trauma. Currently, the method of grading pancreatic injury is performed through radiological techniques such as CT or US, or through confirmation during surgery, and AAST is a commonly used classification system. A limitation of such a classification system is that it is difficult to categorize pancreatic injury, which results in elevated serum pancreatic enzymes, suggesting pancreatic damage without no significant structural injury [15].

In particular, an additional important consideration is that the consequences of pancreatic injury may be worsened by delays in diagnosis, as reported in several studies on pancreatic injury in adults [1,16]. When the diagnosis of pancreatic injury is delayed, morbidity and mortality are known to be 30~40% and 9–34%, respectively [5]. Therefore, it is important to properly diagnose obscure pancreatic injury early.

Serum pancreatic enzymes such as amylase or lipase are not very effective immediately after trauma, but their continuous measurement can be a screening method that helps to diagnose pancreatic injury. However, this is not directly correlated with pancreatic duct injury and thus cannot be generally used to determine the injury grade. In addition, it is difficult to diagnose pancreatic injury with serum pancreatic enzymes alone due to a low sensitivity and specificity, and thus additional radiological diagnosis is required. In fact, there have been cases in which pancreatic enzymes were measured in the normal range even with pancreatic injury in some studies [4,6,17,18,19].

CT is a fast, readily available primary imaging test for evaluating abdominal trauma patients. However, it is difficult to diagnose pancreatic duct injury in children due to its poor sensitivity of about 43~60%. Therefore, it is limited to use as a decision-making tool in the case of pancreatic duct injury [18,20,21].

The value of magnetic resonance cholangiopancreatography (MRCP) in trauma was first reported in 2000 by Fulcher et al. [22]. MRCP is known to be more useful in confirming the integrity of pancreatic duct than CT, but has the disadvantage that it has no therapeutic possibility and an inability to detect non-dilated duct injury. Additionally, it is difficult to use it as a primary imaging test in evaluating abdominal trauma patients, and data on its usefulness in pediatric pancreatic trauma are limited. In particular, in one study of pediatric pancreatic injury, only 24% of patients could be diagnosed with pancreatic duct disruption using MRCP [20,21,22,23].

More recently, a protocol utilizing secretin-enhanced MRCP (S-MRCP) has been developed to diagnose various pancreatic disorders by increasing the secretion of pancreatic exocrine, inducing temporary pancreatic duct dilation. This method has shown the potential to provide more precise diagnostic information compared to traditional imaging modalities such as CT or standard MRI, and can offer information beyond obstructed or disrupted ducts, which may be an advantage over ERCP [24,25,26]. In a study, while only dilation of the main pancreatic duct was observed on MRI, the S-MRCP revealed evidence of a rupture in the main pancreatic duct [26]. However, there are limitations, such as increased cost and examination time, as well as the need for additional personnel for intravenous infusion [24]. Furthermore, the utility of this method for traumatic pancreatic injury has not been widely established.

US is an initial imaging modality commonly used in abdominal trauma and has a high sensitivity to detect intra-abdominal hemorrhage, but its sensitivity and specificity are poor in diagnosing pancreatic injury. It is also less useful than CT in confirming peripancreatic fluid collection and pancreatic enlargement [17,23].

ERCP is an important test for the diagnosis and treatment of biliary tract and pancreatic diseases. Unlike the rich experience of more than 20 years in treating adults, cases of applying this procedure in cases such as sclerosing cholangitis, biliary stones, pancreas divisum, biliary atresia, choledochal cyst, and AUBPD have been rarely reported in children until recently [9]. In traumatic pancreatic injury, the use of ERCP was first reported in 1986 by Hall et al., and several studies have reported the usefulness of ERCP [27].

ERCP is regarded as a gold standard for diagnosing pancreatic duct injury since the specificity is very high, and it has the advantage that therapeutic intervention can be performed simultaneously. Therefore, ERCP has the advantage of avoiding surgery with the potential risk of splenectomy, and even if surgery is necessary it enables surgeons to quickly determine that surgical treatment is necessary, and to determine the appropriate type of surgery by locating the exact injured area. However, it is invasive and may not be readily available in all hospitals. In addition, it is difficult to evaluate the pancreatic parenchyma and surrounding tissue, and in the case of pancreatic duct obstruction, the evaluation of the distal part may be impossible [18,20,21,23,28]. In pediatric patients, there are specific concerns regarding ERCP, including the technical challenges of cannulating the smaller ampulla, peri-pancreatic infections, and the occurrence of post-ERCP pancreatitis with the potential for the development of severe pancreatitis [29]. Although several studies have reported that ERCP is safe and effective in diagnosing pancreatic duct injury in pediatric patients, data are limited to retrospective, small, and single center studies [18,20,21,23,28].

In this study, there were no statistically significant differences in basic characteristics other than the time to diet between the ERCP group and the non-ERCP group. It is assumed that the statistical difference in the time to diet was due to fasting before and after the ERCP. In the six (50%) patients of the early ERCP group, ERCP was used to perform therapeutic interventions or used as a decision-making tool for surgery. In the two (66.6%) patients of the late ERCP group, ERCP could be used to resolve the pancreas-related complication after surgery, and in the one other patient, it was also used to resolve pancreas-related complication that occurred during conservative management. Therefore, in nine out of fifteen patients in the ERCP group, ERCP was able to determine the direction of treatment or help in the treatment process, and in the remaining six patients, conservative treatment could also be determined.

Treatment strategies for pancreatic injuries are largely determined by the severity and location of the injury; however, there has been controversy in pediatric pancreatic injury to date. While nonoperative management is widely accepted as the main treatment strategy in grade 1 and 2 injuries, there is a disagreement between early operative management and nonoperative management in grade 3 or higher injuries accompanied by pancreatic duct injury in several studies [3,17,29,30,31]. In this study, out of fourteen pancreatic injuries of grade 3 or higher, five patients underwent pancreas-related surgery, and the remaining nine patients underwent nonoperative management, but there were no statistically significant differences in clinical outcomes other than ICU stay days. There was one death in the operative management group, but it was due to brain death, not pancreas-related death. Strangely, a paradoxical result was reported in the operative management group of this study, with high ISS but low PTS. We presumed that the reason for this was that ISS mainly evaluates the degree of injury to each part of the body based on abbreviated injury scale (AIS), whereas PTS evaluates the overall condition of the body based on weight, airway status, systolic blood pressure, mental status, skeletal injury, and cutaneous wounds. In addition, statistically more abdominal co-injury was found in the operative management group. Surgery may be necessary if there is abdominal co-injury, but if there is only pancreatic injury it may be able to avoid surgery through therapeutic intervention via ERCP, or to determine the appropriate type of surgery in advance.

Interestingly, when analyzing the patient group by dividing it into two age groups (10 years and younger, and older than 10 years) to examine the differences in injury mechanisms, it was found that blunt force was the most common mechanism in both groups, accounting for 50% and 36.8%, respectively. In the group aged 10 years and younger, passenger traffic accidents and pedestrian traffic accidents accounted for a significant portion, at 41.7%. In the group older than 10 years, motorcycle traffic accidents and falling down comprised a substantial portion, at 42.2%. According to the study on pediatric pancreatic injury over a 10-year period, injuries caused by vehicles such as motor vehicles, motorcycles, and bicycles accounted for approximately 75% of cases [2]. Considering the prevalence of injuries caused by vehicles, despite regional and environmental differences, being aware of these injury mechanisms can help in developing effective preventive measures.

In this study, the final, radiologic, and ERCP injury grades were compared, respectively. When comparing the radiologic injury grade and the final injury grade, the final grade was significantly higher, but there was no statistically significant difference between the ERCP injury grade and the final injury grade. Overall, the final injury grade was often higher than the radiologic grade, showing a significant difference. Therefore, it may be difficult to confirm the pancreatic injury grade by radiology alone, so ERCP may be needed.

When ERCP is performed on children, general anesthesia is generally used more frequently than intravenous anesthesia because of a greater concern about respiratory complications due to greater airway resistance and more frequent airway obstruction compared to in adults [10]. In this study, however, intravenous anesthesia was used for most of the patients, and no anesthesia-related complications occurred. Therefore, it is thought that ERCP can be safely performed through intravenous anesthesia in children, but further studies on anesthesia will be needed.

This study had several limitations. First, there may be a selection bias because it is a retrospective design. Second, it is difficult to confirm the usefulness of ERCP in pediatric pancreatic injury with this study alone because of the single institution and the small sample size. Third, since this study was conducted on patients recruited for a long period, there may be changes in the study data themselves due to the development of radiology and ERCP technology and equipment, and changes in the skills of ERCP performers. Lastly, it was difficult to determine the usefulness of serum pancreatic enzymes and inflammatory markers due to challenges in checking serial laboratory tests. Additionally, the difficulty in conducting MRI in trauma patients due to the health insurance policies in our country resulted in a lack of patients who underwent MRCP or S-MRCP. Therefore, the effectiveness of these imaging modalities could not be evaluated. Despite these limitations, this study may be meaningful in its rarity because it is a study on the usefulness of ERCP in uncommon traumatic pancreatic injuries in children.

## 5. Conclusions

ERCP can be safely and usefully performed in the diagnosis and treatment of traumatic pancreatic injuries in children, and it is necessary to consider actively performing ERCP in cases of ambiguous pancreatic injuries.

## Figures and Tables

**Figure 1 diagnostics-13-02044-f001:**
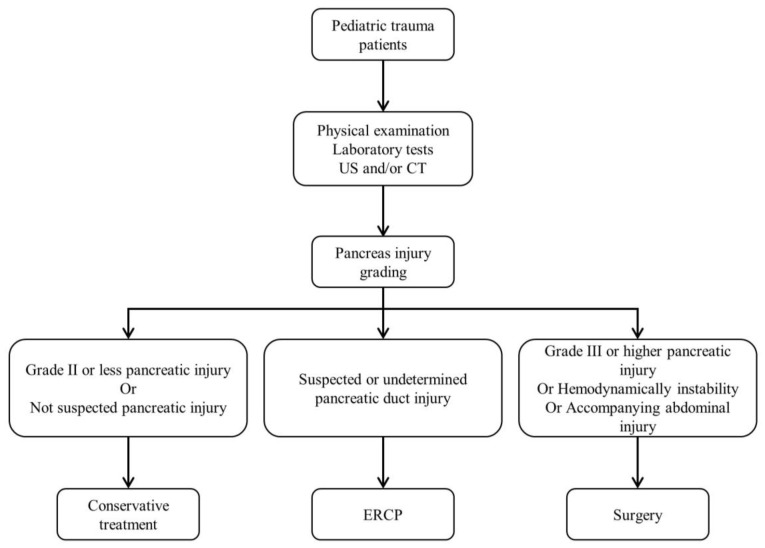
Diagnostic and treatment algorithms for traumatic pancreatic injury patients in children. US: ultrasonography, CT: computed tomography, ERCP: endoscopic retrograde cholangiopancreatography.

**Figure 2 diagnostics-13-02044-f002:**
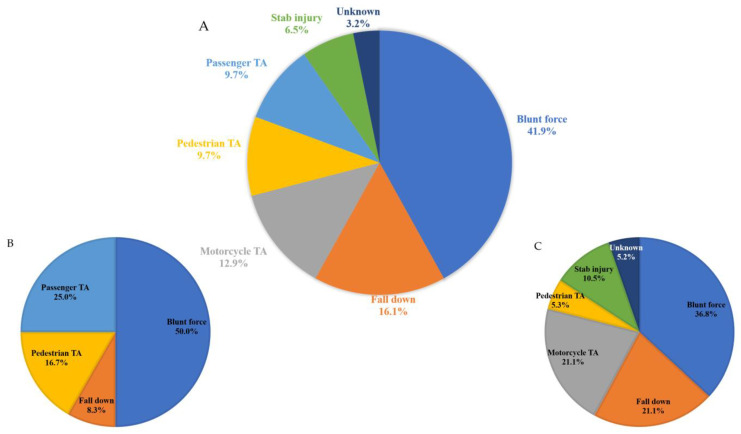
Mechanisms of pancreatic injury. (**A**) Pediatric patients under the age of 18, (**B**) pediatric patients aged 10 years and younger, and (**C**) pediatric patients older than 10 years. TA: traffic accident.

**Figure 3 diagnostics-13-02044-f003:**
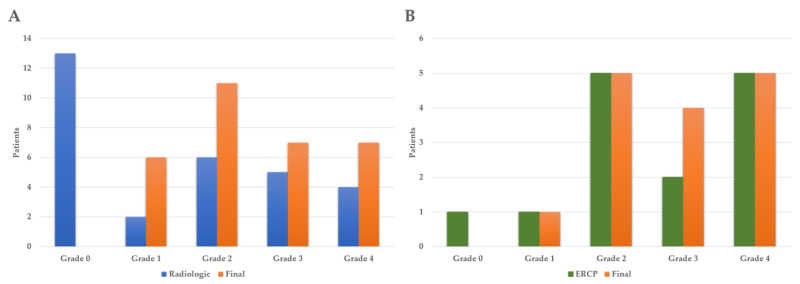
Comparison of radiologic, ERCP, and final injury grade. (**A**) Comparison of radiologic with final injury grade (*p* < 0.001), (**B**) comparison of ERCP with final injury grade (*p* = 0.190). ERCP: endoscopic retrograde cholangiopancreatography.

**Table 1 diagnostics-13-02044-t001:** AAST Pancreas injury scale.

	Grade	Injury Description
I	Hematoma	Minor contusion without ductal injury
	Laceration	Superficial laceration without ductal injury
II	Hematoma	Major contusion without ductal injury or tissue loss
	Laceration	Major laceration without ductal injury or tissue loss
III	Laceration	Distal transection or pancreatic parenchymal injury with ductal injury
IV	Laceration	Proximal transection or pancreatic parenchymal injury involving the ampulla
V	Laceration	Massive disruption of the pancreatic head

AAST: the American Association for the Surgery of Trauma.

**Table 2 diagnostics-13-02044-t002:** Clinical characteristics.

Variables	*n* = 31 (%)
Age (year)	11.71 ± 4.86
Sex (male)	21 (67.7)
Weight (kg)	42.18 ± 18.96
Systolic blood pressure (mmHg)	113.23 ± 16.06
Diastolic blood pressure (mmHg)	67.39 ± 12.34
Initial shock	5 (16.1)
Injury mechanism	
Blunt force	13 (41.9)
Falling down	5 (16.1)
Motorcycle TA	4 (12.9)
Pedestrian TA	3 (9.7)
Passenger TA	3 (9.7)
Stab injury	2 (6.5)
Unknown	1 (3.2)
Abdominal co-injury	16 (51.6)
AIS1	1.19 ± 1.52
AIS2	0.39 ± 0.72
AIS3	0.81 ± 1.33
AIS4	4.16 ± 1.07
AIS5	0.68 ± 1.38
AIS6	1.16 ± 0.52
ISS	27.32 ± 10.99
PTS	9.23 ± 1.88
Amylase (U/L)	582.69 ± 673.00
Lipase (U/L)	3375.55 ± 6662.45
CRP (mg/dL)	2.22 ± 3.72
DNI (%)	1.93 ± 2.24
Pancreatic injury site	
Head	5 (16.1)
Neck	3 (9.7)
Body	10 (32.3)
Tail	8 (25.8)
2 or more	5 (16.1)
CT	29 (93.5)
US	16 (51.6)
ERCP	15 (48.4)
Operation	10 (32.3)
Injury grade	
I	6 (19.4)
II	11 (35.5)
III	7 (22.6)
IV	7 (22.6)
V	0 (0.0)
Early complications	15 (48.4)
Late complications	6 (19.4)
Time to diet (day)	9.04 ± 5.55
Time to parenteral nutrition (day)	2.32 ± 1.57
ICU stay (day)	2.65 ± 3.42
Hospital stay (day)	30.97 ± 59.41
Mortality	2 (6.5)

TA: traffic accident, AIS: Abbreviated Injury Scale, ISS: Injury Severity Score, PTS: Pediatric Trauma Score, CRP: C-reactive protein, DNI: delta neutrophil index, CT: computed tomography, US: ultrasonography, ERCP: endoscopic retrograde cholangiopancreatography, ICU: intensive care unit.

**Table 3 diagnostics-13-02044-t003:** Comparison between operation and non-operation groups.

Variables	Operation (*n* = 10)	Non-Operation (*n* = 21)	*p*-Value
Age (year)	12.30 ± 4.88	11.43 ± 4.95	0.649
Sex (male)	7 (70.0)	14 (66.7)	1.000 *
Weight (kg)	42.40 ± 15.49	42.07 ± 20.77	0.965
Systolic blood pressure (mmHg)	108.30 ± 16.86	115.57 ± 15.52	0.245
Diastolic blood pressure (mmHg)	66.90 ± 12.69	67.62 ± 12.48	0.882
Initial shock	4 (40.0)	1 (4.8)	0.027 *
Injury mechanism			0.101 *
Blunt force	3 (30.0)	10 (47.6)	
Falling down	0 (0.0)	5 (23.8)	
Motorcycle TA	2 (20.0)	2 (9.5)	
Pedestrian TA	1 (10.0)	2 (9.5)	
Passenger TA	1 (10.0)	2 (9.5)	
Stab injury	2 (20.0)	0 (0.0)	
Unknown	1 (10.0)	0 (0.0)	
Abdominal co-injury	8 (80.0)	8 (38.1)	0.054 *
ISS	35.20 ± 9.56	23.57 ± 9.70	0.004
PTS	7.90 ± 2.13	9.86 ± 1.39	0.005
Amylase (U/L)	386.00 ± 520.73	676.35 ± 727.20	0.269
Lipase (U/L)	6502.40 ± 13,158.57	2333.27 ± 2472.75	0.519
CRP (mg/dL)	2.87 ± 5.11	1.99 ± 3.37	0.699
DNI (%)	1.63 ± 1.70	2.02 ± 2.46	0.806
Pancreatic injury site			0.084 *
Head	5 (50.0)	2 (9.5)	
Neck	0 (0.0)	1 (4.8)	
Body	1 (10.0)	9 (42.9)	
Tail	2 (20.0)	6 (28.6)	
2 or more	2 (20.0)	3 (14.3)	
CT	8 (80.0)	21 (100.0)	0.097 *
US	4 (40.0)	12 (57.1)	0.458 *
ERCP	4 (40.0)	11 (52.4)	0.704 *
Injury grade			0.307 *
I	0 (0.0)	6 (28.6)	
II	4 (40.0)	7 (33.3)	
III	3 (30.0)	4 (19.0)	
IV	3 (30.0)	4 (19.0)	
V	0 (0.0)	0 (0.0)	
Early complications	3 (30.0)	12 (57.1)	0.252 *
Late complications	2 (20.0)	4 (19.0)	1.000 *
Time to diet (day)	9.88 ± 5.03	8.68 ± 5.85	0.620
Time to parenteral nutrition (day)	2.22 ± 1.39	2.38 ± 1.71	0.821
ICU stay (day)	4.60 ± 3.84	1.71 ± 2.85	0.025
Hospital stay (day)	56.00 ± 101.90	19.05 ± 12.57	0.106
Mortality	2 (20.0)	0 (0.0)	0.097 *

TA: traffic accident, ISS: Injury Severity Score, PTS: Pediatric Trauma Score, CRP: C-reactive protein, DNI: delta neutrophil index, CT: computed tomography, US: ultrasonography, ERCP: endoscopic retrograde cholangiopancreatography, ICU: intensive care unit. * Result of Fisher’s exact test.

**Table 4 diagnostics-13-02044-t004:** Comparison between ERCP and non-ERCP groups.

Variables	ERCP (*n* = 15)	Non-ERCP (*n* = 16)	*p*-Value
Age (year)	12.80 ± 4.36	10.69 ± 5.21	0.233
Sex (male)	11 (73.3)	10 (62.5)	0.704 *
Weight (kg)	45.30 ± 17.07	39.25 ± 20.69	0.384
Systolic blood pressure (mmHg)	117.47 ± 10.25	109.25 ± 19.56	0.158
Diastolic blood pressure (mmHg)	68.33 ± 15.02	66.50 ± 9.61	0.687
Initial shock	1 (6.7)	4 (25.0)	0.333 *
Injury mechanism			1.000 *
Blunt force	7 (46.7)	6 (37.5)	
Falling down	2 (13.3)	3 (18.8)	
Motorcycle TA	2 (13.3)	2 (12.5)	
Pedestrian TA	1 (6.7)	2 (12.5)	
Passenger TA	1 (6.7)	2 (12.5)	
Stab injury	1 (6.7)	1 (6.3)	
Unknown	1 (6.7)	0 (0.0)	
Abdominal co-injury	6 (40.0)	10 (62.5)	0.289
ISS	25.87 ± 8.30	28.69 ± 13.15	0.484
PTS	9.73 ± 1.71	8.75 ± 1.95	0.147
Amylase (U/L)	706.47 ± 671.17	466.64 ± 675.08	0.330
Lipase (U/L)	6115.22 ± 9421.55	1134.00 ± 1021.19	0.152
CRP (mg/dL)	2.72 ± 4.18	1.78 ± 3.50	0.643
DNI (%)	0.93 ± 1.35	2.79 ± 2.59	0.144
Pancreatic injury site			0.870 *
Head	3 (20.0)	4 (25.0)	
Neck	1 (6.7)	0 (0.0)	
Body	5 (33.3)	5 (31.3)	
Tail	3 (20.0)	5 (31.3)	
2 or more	3 (20.0)	2 (12.5)	
CT	14 (93.3)	15 (93.8)	1.000 *
US	9 (60.0)	7 (43.8)	0.366
Operation	4 (26.7)	6 (37.5)	0.704 *
Injury grade			0.296 *
I	1 (6.7)	5 (31.3)	
II	5 (33.3)	6 (37.5)	
III	4 (26.7)	3 (18.8)	
IV	5 (33.3)	2 (12.5)	
V	0 (0.0)	0 (0.0)	
Early complications	10 (66.7)	5 (31.3)	0.076
Late complications	4 (26.7)	2 (12.5)	0.394 *
Time to diet (day)	10.93 ± 5.89	6.67 ± 4.21	0.045
Time to parenteral nutrition (day)	2.46 ± 1.56	2.17 ± 1.64	0.650
ICU stay (day)	1.67 ± 2.53	3.56 ± 3.95	0.125
Hospital stay (day)	24.33 ± 13.46	37.19 ± 82.48	0.556
Mortality	0 (0.0)	2 (12.5)	0.484 *

TA: traffic accident, ISS: Injury Severity Score, PTS: Pediatric Trauma Score, CRP: C-reactive protein, DNI: delta neutrophil index, CT: computed tomography, US: ultrasonography, ERCP: endoscopic retrograde cholangiopancreatography, ICU: intensive care unit. * Result of Fisher’s exact test.

**Table 5 diagnostics-13-02044-t005:** Clinical characteristics of patients who underwent ERCP.

Case	Age/Sex	Mechanism of Injury	Pancreatic Injury Site	Rad Grade	ERCP Grade	Injury to ERCP (Days)	ERCP Findings	Intervention	Treatment	Complications	ICU/Hospital Stay Days	Time to Diet/PN
1	15/M	Falling down	Body	2	0	1	Normal	None	Conserv.	Pseudocyst	0/8	4/1
2	14/M	Stab injury	Body	3	2	11 (post-op.)	Irregular ductal change on body of pancreas without definite ductal leakage	EPST + Stent	Surgery followed by ERCP	Post-pancreatitis	4/42	9/1
3	6/M	In car TA	Body	0	1	3	Normal	None	Conserv.	None	0/14	11/6
4	7/M	Bicycle handlebars	Head	0	2	4	Parenchymal staining of head, Not visualized main pancreatic duct	None	Conserv.	Distal pancreatic atrophy due to pancreatic duct stenosis	0/10	7/none
5	8/M	Bicycle handlebars	Head, body–tail	3	4	0	Leakage (head and body-tail)	EPST + Stent	PAD due to pseudocyst	Pseudocyst	0/40	15/2
6	10/M	Bicycle handlebars	Body	0	0	0	Normal	None	Conserv.	None	0/6	4/none
7	11/M	Bicycle handlebars	Head	2	2	4	Normal	None	Surgery for disturbed duodenal passage	None	0/25	18/1
8	12/F	Bicycle handlebars	Tail	3	3	7	Leakage on tail, Pseudocyst	Stent	PAD due to pseudocyst	Pseudocyst	0/24	14/3
9	15/M	Bump into a wall while cycling	Neck	3	3	0	Leakage on neck with main pancreatic duct laceration	Stent	Conserv.	Post-pancreatitis, Chronic pancreatitis	2/32	6/1
10	17/F	Unknown	Head	4	4	15 (post-op.)	Stricture of neck, could not find leakage	EPST + Stent	Surgery followed by ERCP	Pancreatic duct stricture, DM	8/39	14/3
11	17/M	Bicycle handlebars	Head-neck	4	4	2	Leakage of head to neck	EPST + Stent	Conserv.	Pancreatic duct stricture	3/27	6/4
12	17/M	Motorcycle	Tail	0	2	9	No definite ductal leakage	Stent	PAD due to pseudocyst	Pseudocyst	1/27	21/1
13	18/F	Motorcycle	Head-neck	2	4	0	Leakage of head to neck	None	Surgery	IGT	1/16	8/4
14	7/F	Bump under a truck	Body	2	2	1	Could not find leakage due to insufficient dye	EPST	PAD due to pseudocyst	Pseudocyst	6/46	21/2
15	18/M	Fall down	Tail	0	2	2	Acute pancreatitis with diffuse narrowed pancreatic duct	None	Conserv.	Post-pancreatitis	0/9	6/3

ERCP: endoscopic retrograde cholangiopancreatography, ICU: intensive care unit, PN: parenteral nutrition, M: male, TA: traffic accident, EPST: endoscopic pancreatic sphincterotomy, PAD: percutaneous abscess drainage, F: female, DM: diabetes mellitus, IGT: Impaired glucose tolerance.

## Data Availability

The datasets used and/or analyzed during the current study are available from the corresponding author upon reasonable request.

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
