# Peer review of "Usefulness of Endoscopic Retrograde Cholangiopancreatography in the Diagnosis and Treatment of Traumatic Pancreatic Injury in Children"

_diagnostics, 2023, doi:10.3390/diagnostics13122044_

Round 1
Reviewer 1 Report
Thank you for the possibility to review the manuscript titled: “Usefulness of Endoscopic Retrograde Cholangiopancreatography in the Diagnosis and Treatment of Traumatic Pancreatic Injury in Children”. The manuscript is interesting and easy to read. However, ther are several important points, whhic should be considered:
-Please review the language of the manuscript;
-Figure 1 (AAST pancreas injury scale) should be changed to a table;
-Table 5 can be presented in an alternative fashion, as a diagram as it has little information for a table;
-Please describe the diagnostic algorithm. A standard evaluation for abdominal injury would be clinical evaluation + laboratory tests + USG and/CT. A CT scan with iv enhancement is highly informative for pancreatic trauma and in mixed results can be followed by MRCP. Please give more comment and discuss this in the manuscript.
-“Pancreatic injury is reported as 3-12% of blunt abdominal trauma and 1-12% of penetrating injury, and is known as the fourth most common abdominal solid organ injury after the spleen, liver, and kidney [1,2]” This requires specification, as isolated pancreatic trauma is rare in adults and is usually seen as a part of other traumas. Please specify if this information is preset for adults or children.
-Another important thing is that children, although is a collective term implies different age groups and different mechanisms of trauma and type of injuries. This requires more discussion.
-Please discuss MRCP and MRCP+secretine test in the discussion section as this method is highly informative.
-Please discuss post-ERCP pancreatitis as one of the possible complications of the procedure.
Please take into account the recommendations in the spirit of improving the quality of the submission.
Minor editing
Reviewer 2 Report
Dear Authors I've read with interest your work that is well written and addresses a potential tricky situation.
I've only few observations for you:
1. line 34: please rephrase the sentence because "pancreatic injury is the forth most frequent injury AFTER liver, spleen..., not followed by.. (see line 222)
2. line 90: which kind of radiologic imaging do you refer ?
3: the "initial" serum levels of pancreatic enzymes how many time after trauma were checked?
4. can you explain why, in your opinion OIS2 pancreatic injuries were managed more frequently by surgery? (table 2)
Round 2
Reviewer 1 Report
There are no recommendations. The authors have made all of the necessary corrections.
Author Response
Once again, we would like to express our gratitude for your insightful comments. Your valuable input has undoubtedly elevated the worth and caliber of this study. We truly appreciate your sincere review of this study.